# Reactive metabolite production is a targetable liability of glycolytic metabolism in lung cancer

Alba Luengo [1,2], Keene L. Abbott [1,2], Shawn M. Davidson[1,2,3], Aaron M. Hosios[1,2], Brandon Faubert [4], Sze Ham Chan[5], Elizaveta Freinkman[5], Lauren G. Zacharias[4], Thomas P. Mathews[4,6], Clary B. Clish [3], Ralph J. DeBerardinis[4,6,7], Caroline A. Lewis [5] & Matthew G. Vander Heiden [1,2,3,8]*

Increased glucose uptake and metabolism is a prominent phenotype of most cancers, but efforts to clinically target this metabolic alteration have been challenging. Here, we present evidence that lactoylglutathione (LGSH), a byproduct of methylglyoxal detoxification, is elevated in both human and murine non-small cell lung cancers (NSCLC). Methylglyoxal is a reactive metabolite byproduct of glycolysis that reacts non-enzymatically with nucleophiles in cells, including basic amino acids, and reduces cellular fitness. Detoxification of methylglyoxal requires reduced glutathione (GSH), which accumulates to high levels in NSCLC relative to normal lung. Ablation of the methylglyoxal detoxification enzyme glyoxalase I (Glo1) potentiates methylglyoxal sensitivity and reduces tumor growth in mice, arguing that targeting pathways involved in detoxification of reactive metabolites is an approach to exploit the consequences of increased glucose metabolism in cancer.

---

[1] Koch Institute for Integrative Cancer Research, Massachusetts Institute of Technology, Cambridge, MA 02139, USA. [2] Department of Biology, Massachusetts Institute of Technology, Cambridge, MA 02139, USA. [3] Broad Institute of MIT and Harvard University, Cambridge, MA 02142, USA. [4] Children's Medical Center Research Institute, University of Texas Southwestern Medical Center, Dallas, TX, USA. [5] Whitehead Institute for Biomedical Research, Massachusetts Institute of Technology, Cambridge, MA 02142, USA. [6] Howard Hughes Medical Institute, University of Texas Southwestern Medical Center, Dallas, TX, USA. [7] Department of Pediatrics and Eugene McDermott Center for Human Growth and Development, University of Texas Southwestern Medical Center, Dallas, TX, USA. [8] Dana-Farber Cancer Institute, Boston, MA 02115, USA. *email: mvh@mit.edu

Cancer cells engage in altered metabolism to support the biosynthetic demands of malignant proliferation[1,2]. Reprogrammed metabolism distinguishes tumors from the tissues from which they arise, raising the possibility that tumor metabolism could be targeted for cancer therapy. Elevated glucose uptake and flux through glycolysis is a salient metabolic feature of many cancers[3]. However, efforts to target glucose metabolism directly, including inhibitors of glucose uptake, glycolysis, and lactate excretion, have seen limited clinical success, likely because many non-malignant cells also require glucose metabolism[4,5].

Targeting secondary effects of increased glucose metabolism is an alternative approach to exploit this metabolic phenotype in cancer, and synthetic lethality approaches involving glycolysis have been proposed as potential cancer treatments. For example, passenger deletion of one isoform of the glycolytic enzyme enolase can result in increased dependence on the remaining isoform[6,7]. Other indirect consequences of increased glucose metabolism may present liabilities that are differentially important in cancer compared to normal cells and could be applicable to broader subsets of patients.

Here, we characterize secondary effects of reprogrammed glucose metabolism in lung tumors in order to identify potential anticancer targets. We perform untargeted metabolomics on glucose-avid non-small cell lung cancer (NSCLC) mouse models of human cancer and find that accumulation of reduced glutathione is prominent in tumors relative to normal lung. We determine that lung tumors also accumulate the glutathione conjugate lactoylglutathione (LGSH), which is produced by the enzyme glyoxalase I (Glo1) to detoxify methylglyoxal, a reactive metabolite that has also been reported to accumulate in tumors[8,9]. We confirm that methylglyoxal is a byproduct of glucose metabolism in NSCLC and additionally show that Glo1 expression supports tumor growth in mice. Taken together, this study suggests that inhibition of pathways that detoxify reactive metabolites, including methylglyoxal, could be one approach to target the increased glucose metabolism of tumors.

## Results

### Reduced glutathione accumulates in NSCLC.
Increased glucose metabolism in cancer results in different steady-state concentrations of metabolites in tumors relative to normal tissues, which can have implications for cell signaling, bioenergetics, and the availability of substrates that support biosynthesis and cell proliferation[10,11]. To identify metabolites that differentially accumulate in lung cancers relative to normal lung tissue, we performed untargeted LCMS-based metabolomics on normal lung and lung tumor tissue derived from two autochthonous mouse models of Kras$^{G12D}$-driven NSCLC that have been shown to avidly consume glucose[12–14]. The Kras$^{G12D}$ (LA2) model, which is initiated by spontaneous recombination of a latent allele, resulting in mutant Kras expression[15], and a second model, which involves Cre-mediated activation of Kras$^{G12D}$ expression and deletion of Trp53 (KP), form lung tumors of different grades and invasiveness[16]. Though the majority of metabolites do not differentially accumulate in tumor tissue compared to normal lung, we determined that a subset of metabolites is enriched in NSCLC tumors including reduced glutathione (GSH) (Supplementary Fig. 1a).

GSH is a critical cofactor cells employ to detoxify reactive molecules, including a number of oxygen-containing chemical species, termed reactive oxygen species (ROS), that are produced as a byproduct of mitochondrial metabolism and other cellular processes[17]. Increased ROS generation is associated with malignancy, and higher GSH levels in tumors are proposed to benefit cancer cells by enabling greater ROS detoxification[18,19].

Cells neutralize ROS via the action of various enzyme complexes, some of which ultimately result in the oxidation of reduced GSH to the oxidized form of glutathione, GSSG[20]. Thus, we reasoned that if glutathione levels are elevated in lung cancer as a response to increased oxidative stress, the increase in GSH might be accompanied by an increase in GSSG. However, although we confirmed that GSH was significantly elevated in lung tumors from both models, levels of GSSG were increased only in tumors arising from the LA2 model, and not in the more aggressive KP model which also displayed higher levels of GSH (Fig. 1a). Furthermore, the relative fold increase in GSH was greater than that of GSSG in both models, suggesting that lung cancers may have a more reduced GSH/GSSG ratio than normal lung tissue. This argues against oxidative stress as the only explanation for GSH elevation in these cancers, and that elevated GSH may serve an additional function beyond ROS detoxification in lung cancer.

### Increased methylglyoxal detoxification in NSCLC.
GSH has a nucleophilic cysteine sulfhydryl group that permits glutathione to react with electrophiles and form GSH-conjugates. The formation of GSH-conjugates is involved in electrophile detoxification, and thus higher levels of glutathione might prevent these reactive metabolites from reacting with and damaging cellular macromolecules[21]. To explore this possibility, we queried the untargeted metabolomics data acquired from normal lung and lung tumors to determine whether known glutathione-electrophile conjugates were detected. Consistent with a role for GSH in detoxifying electrophiles in tumors, we identified metabolites that were increased in tumors relative to normal lung with masses corresponding to glutathione-electrophile conjugates (Supplementary Fig. 1b). Among these was LGSH, which is the conjugate of methylglyoxal and GSH (Fig. 1b).

To validate that LGSH was elevated in NSCLC tumors, we analyzed metabolites from lung allografts generated from KP lung tumor cells. In this analysis, we compared the MS/MS spectrum of an LGSH standard to that from lung tumors, which together with retention time confirmation, permitted validation of peak identity (Supplementary Fig. 1c, d). Next, using targeted metabolomics, we determined that relative to normal lung, LGSH was elevated in tumors arising in LA2 and KP mice (Fig. 1c) and in KP lung tumor allografts (Fig. 1d). Furthermore, LGSH levels were elevated in many human NSCLC tumors relative to adjacent non-cancerous lung (Fig. 1e, Supplementary Fig. 1e), suggesting that LGSH elevation is also a feature of human NSCLC.

LGSH accumulation is of interest because methylglyoxal is known to be a byproduct of increased glucose metabolism associated with hyperglycemia and diabetes[22], and can contribute to vascular damage and other complications arising from increased circulating glucose levels[23–26]. Methylglyoxal has also been reported to accumulate in cancers[8,9]. At high levels, methylglyoxal induces cytotoxicity because it potently and irreversibly reacts with proteins and nucleic acids to form stable adducts referred to as advanced glycation end-products (AGEs)[27]. In order to prevent AGE formation, the enzyme Glo1 is expressed across tissues[28] and catalyzes the covalent conjugation of methylglyoxal with GSH to form LGSH (Fig. 1b). Methylglyoxal reacts non-enzymatically with GSH to form a hemithioacetal intermediate; however, this process is reversible, and Glo1 catalyzes the isomerization of the hemithioacetal intermediate to produce LGSH, which is more stable. In this way, Glo1 acts to irreversibly sequester reactive methylglyoxal and prevent the non-specific reaction of this electrophile with macromolecules to damage cells[28]. We assessed Glo1 expression, and found that Glo1 was increased in KP lung tumors relative to normal lung tissues (Fig. 1f). Together, these data suggest that lung tumor

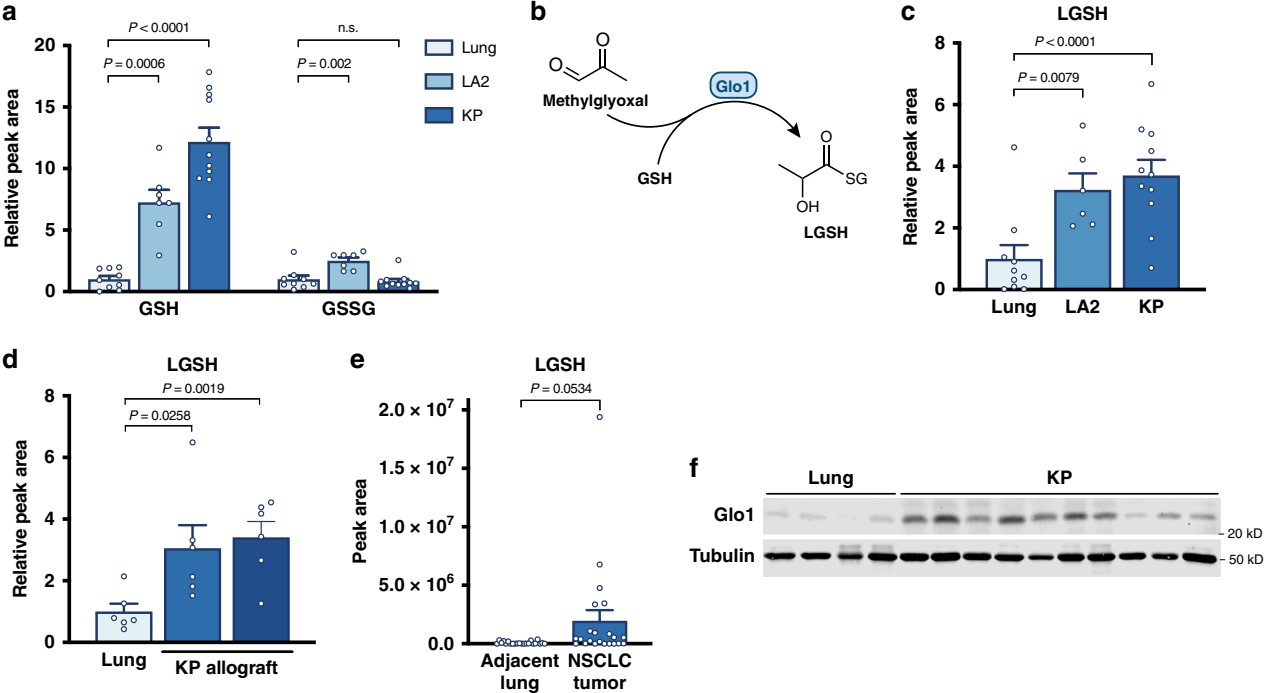

**Fig. 1 Reduced glutathione and lactoylglutathione accumulate in NSCLC compared to normal lung tissue. a** Relative levels of reduced glutathione (GSH) and oxidized glutathione (GSSG) in extracts from normal mouse lung tissue (Lung), or from tumors arising in LA2 and KP mouse NSCLC models as measured by targeted LCMS (Lung, $n = 9$; LA2, $n = 6$; KP, $n = 11$). **b** Schematic depicting the reaction catalyzed by glyoxalase I (Glo1) to conjugate methylglyoxal to reduced glutathione (GSH) to produce lactoylglutathione (LGSH). **c** Relative levels of LGSH in normal mouse lung tissue, LA2 tumors, and KP tumors as assessed by LCMS. Data are normalized to tissue weight and peak area (Lung, $n = 9$; LA2, $n = 6$; KP, $n = 11$). **d** Quantification of LGSH in normal lung tissue and in KP allografts by LCMS. The allografts had been generated from two independent cell lines that were derived from a KP mouse lung tumor[13] ($n = 6$). **e** LGSH metabolite levels as determined by LCMS detected in extracts from lung cancer and adjacent non-cancer lung tissue resected from 22 human patients with NSCLC. The LGSH peak area for each paired lung cancer and adjacent non-cancer patient sample is shown in Supplementary Fig. 1e. **f** Western blot analysis to assess Glo1 expression in normal mouse lung tissue and in KP lung tumors. $P$ values were calculated by unpaired, two-tailed $t$-test. Values shown in panels **a**, **c**, **d**, and **e** denote the mean ± SEM.

tissues have increased Glo1 expression and production of LGSH to detoxify methylglyoxal.

**Methylglyoxal is produced from glucose in NSCLC.** One mechanism of methylglyoxal production is non-enzymatic formation from the degradation of triose-phosphate intermediates of glycolysis[27,29]. In this case, a major source of carbon for methylglyoxal formation is glucose, as is observed in hyperglycemia[22]. However, methylglyoxal can also be generated as a byproduct of threonine and glycine interconversion[30–32], and production of methylglyoxal from glycine has also been reported in some cancer contexts[33]. Stable isotope experiments can be used to discern the source of methylglyoxal, as conversion of [U-$^{13}$C$_2$]glycine to methylglyoxal results in methylglyoxal with one labeled carbon (m + 1) (Fig. 2a), while methylglyoxal generation from [U-$^{13}$C$_6$]glucose produces fully labeled (m + 3) methylglyoxal (Fig. 2b). We cultured lung cancer cells derived from KP lung tumors in media containing [U-$^{13}$C$_2$]glycine or [U-$^{13}$C$_6$]glucose and measured label incorporation into methylglyoxal using LCMS. Though the reactive nature of methylglyoxal makes reliable direct detection by LCMS difficult, methylglyoxal $^{13}$C labeling can be calculated from the isotope distribution of glutathione and LGSH (Methods). Lung cancer cells cultured with [U-$^{13}$C$_2$]glycine only had ~5% of methylglyoxal m + 1 labeled, despite the fact that 60% of the intracellular glycine pool was labeled under these conditions (Fig. 2c, Supplementary Fig. 2a–c). Conversely, when lung cancer cells were cultured with [U-$^{13}$C$_6$]glucose, the LGSH labeling was consistent with virtually all of the methylglyoxal being m + 3 labeled, matching the labeling pattern

of the triose-phosphate glycolytic intermediate dihydroxyacetone phosphate (Fig. 2d, Supplementary Fig. 2d–f). These results indicate that for lung cancer cells in culture, glucose is the major source of endogenous methylglyoxal.

To determine whether methylglyoxal is also derived from glucose in lung tumors, we assessed the contribution of [U-$^{13}$C$_6$]glucose to methylglyoxal in NSCLC arising in LA2 mice[13]. We infused [U-$^{13}$C$_6$]glucose into the jugular veins of conscious, unrestrained mice for 6 hours, harvested tumors, and analyzed metabolite labeling in tissue by LCMS. The majority of methylglyoxal was m + 3 labeled in tumors, as inferred from LGSH labeling following $^{13}$C glucose infusion (Fig. 2e, Supplementary Fig. 2g, h). These data suggest that methylglyoxal in lung tumors is generated from glucose, consistent with results observed in lung cancer cells in culture.

To confirm that increased methylglyoxal production in lung cancer cells is a direct consequence of increased glycolysis, we assessed whether levels of LGSH track with the extent of glucose metabolism. We exposed lung cancer cells to increasing concentrations of 2-deoxyglucose (2DG), a competitive inhibitor of hexokinase than impairs glucose uptake[34], and found that 2DG reduced LGSH levels in lung cancer cells (Fig. 2f). This further suggests that methylglyoxal production is a consequence of glucose metabolism in cells.

NSCLC arising in these mouse models are known to be 2-deoxy-2-[$^{18}$F]fluoro-D-glucose-positron emission tomography (FDG-PET) positive;[12] however, human lung cancers can exhibit variable levels of glucose uptake[35]. To assess whether methylglyoxal detoxification correlate with increased glucose metabolism in tumors, we evaluated LGSH levels in human tumors with a

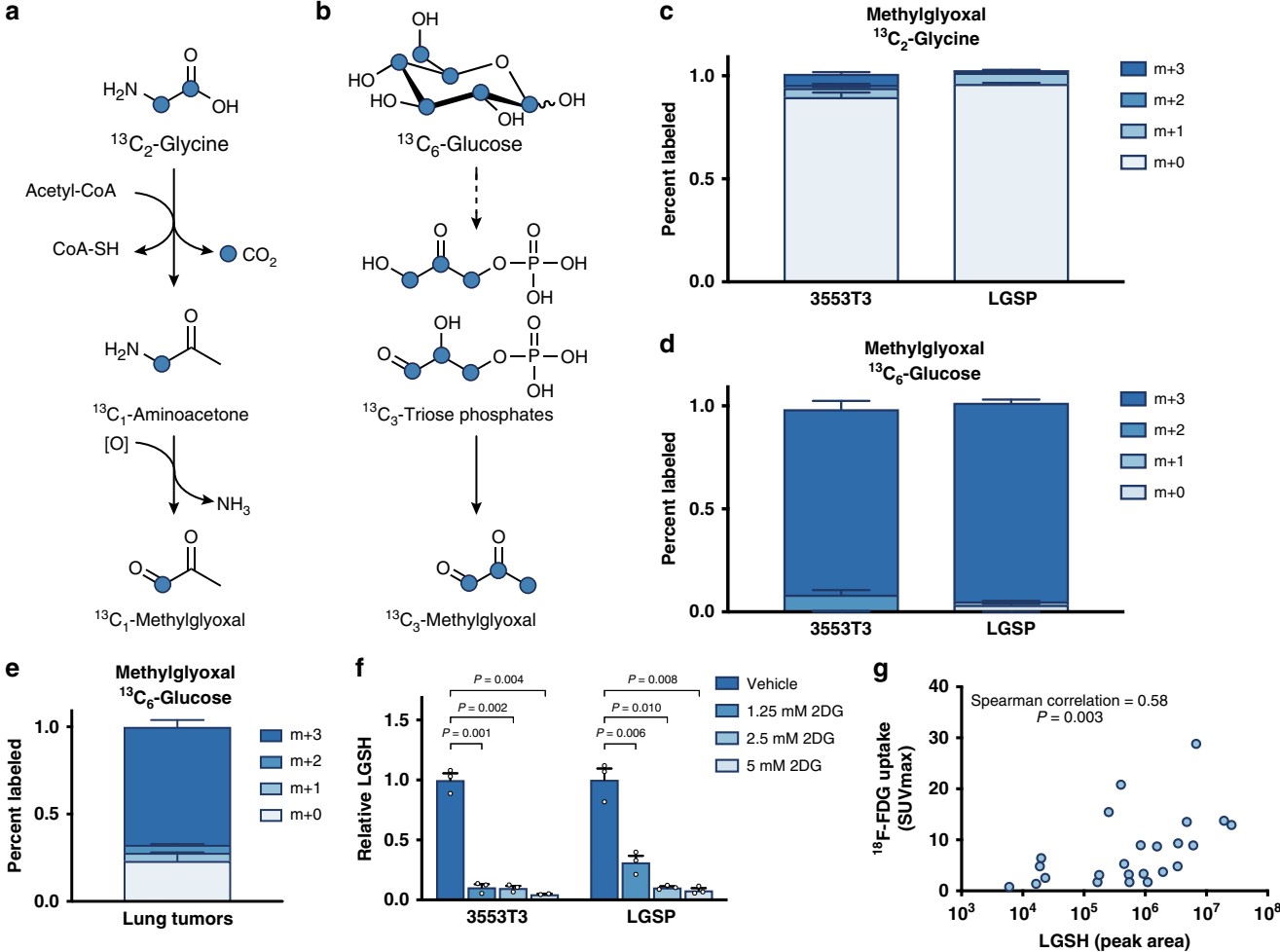

**Fig. 2 Methylglyoxal is produced as a byproduct of glycolysis in NSCLC. a, b** Schematic detailing how labeled carbon (denoted by blue circles) from [U-$^{13}C_2$]glycine (**a**) or [U-$^{13}C_6$]glucose (**b**) can be incorporated into methylglyoxal based on known pathways. **c, d** Isotopomer distribution of methylglyoxal in lung cancer cell lines (3553T3 and LGSP) independently derived from KP mice that were cultured for 24 h in the presence of [U-$^{13}C_2$]glycine (**c**) or [U-$^{13}C_6$]glucose (**d**). The isotopomer distribution was calculated using data shown in Supplementary Fig. 2a-b;d-e ($n = 3$). **e** Isotopomer distribution of methylglyoxal in autochthonous lung tumors arising in LA2 mice following a 6 h infusion of 30 mg/kg/min [U-$^{13}C_6$]glucose. Values were normalized to plasma enrichment of glucose and the isotopomer distribution was calculated using data shown in Supplementary Fig. 2g, h ($n = 8$). **f** Relative lactoylglutathione (LGSH) levels in 3553T3 and LGSP cells as detected by LCMS. Levels are shown for cells cultured in standard media, or media containing the indicated amount of 2-deoxyglucose (2DG) for 24 h. Values shown are the peak area of LGSH normalized to the peak area of reduced glutathione ($n = 3$). $P$ values were calculated by unpaired, two-tailed $t$-test. **g** Correlation between LGSH abundance in samples derived from NSCLC patient samples as measured by LCMS, and the maximum standardized uptake value of (SUVmax) as measured by $^{18}$F-FDG-PET imaging ($n = 24$). Values in **c-f** denote mean ± SEM.

range of known FDG-PET standard uptake values. We found a positive correlation (Fig. 2g), which further supports that increased glucose metabolism in lung cancer tumors contributes to methylglyoxal production and LGSH accumulation.

**Glutathione depletion sensitizes NSCLC to methylglyoxal.** As methylglyoxal production is a byproduct of glucose metabolism, this raises the possibility that the increased accumulation of glutathione observed in tumors at least in part supports methylglyoxal detoxification. To test this hypothesis, we first examined whether depletion of intracellular glutathione could alter the proliferation of NSCLC cells cultured in the presence of exogenous methylglyoxal. Indeed, treatment of lung cancer cells with buthionine sulfoximine (BSO), a compound that inhibits glutathione synthesis[36], rendered cells exquisitely sensitive to exogenous methylglyoxal (Fig. 3a). Erastin, which is a selective inhibitor of system $x_c^-$[37], and limits cystine uptake and cysteine-

dependent glutathione synthesis, also increased lung cancer cell sensitivity to methylglyoxal (Fig. 3b), although menadione, which generates ROS by futile redox cycling[38,39], increased sensitivity of LGSP but not 3553T3 cells to methylglyoxal (Fig. 3c). All three compounds decreased intracellular GSH pools, although menadione did so with slower kinetics (Supplementary Fig. 3a). These data suggest that glutathione depletion, rather than increased ROS generation, has a stronger effect on promoting methylglyoxal sensitivity.

Since GSH is required for Glo1 to detoxify methylglyoxal, depletion of GSH could result in elevation of methylglyoxal levels and accumulation of glycated macromolecules. One adduct formed by non-enzymatic methylglyoxal glycation is the hydroimidazolone MG-H1, which is formed by modification of the arginine guanidino group (Fig. 3d)[40–42]. We found that in the presence of exogenous methylglyoxal, BSO, and erastin increased levels of proteins with MG-H1 adducts (Fig. 3e, f), suggesting that glutathione depletion prevents efficient methylglyoxal detoxification and leads to increased

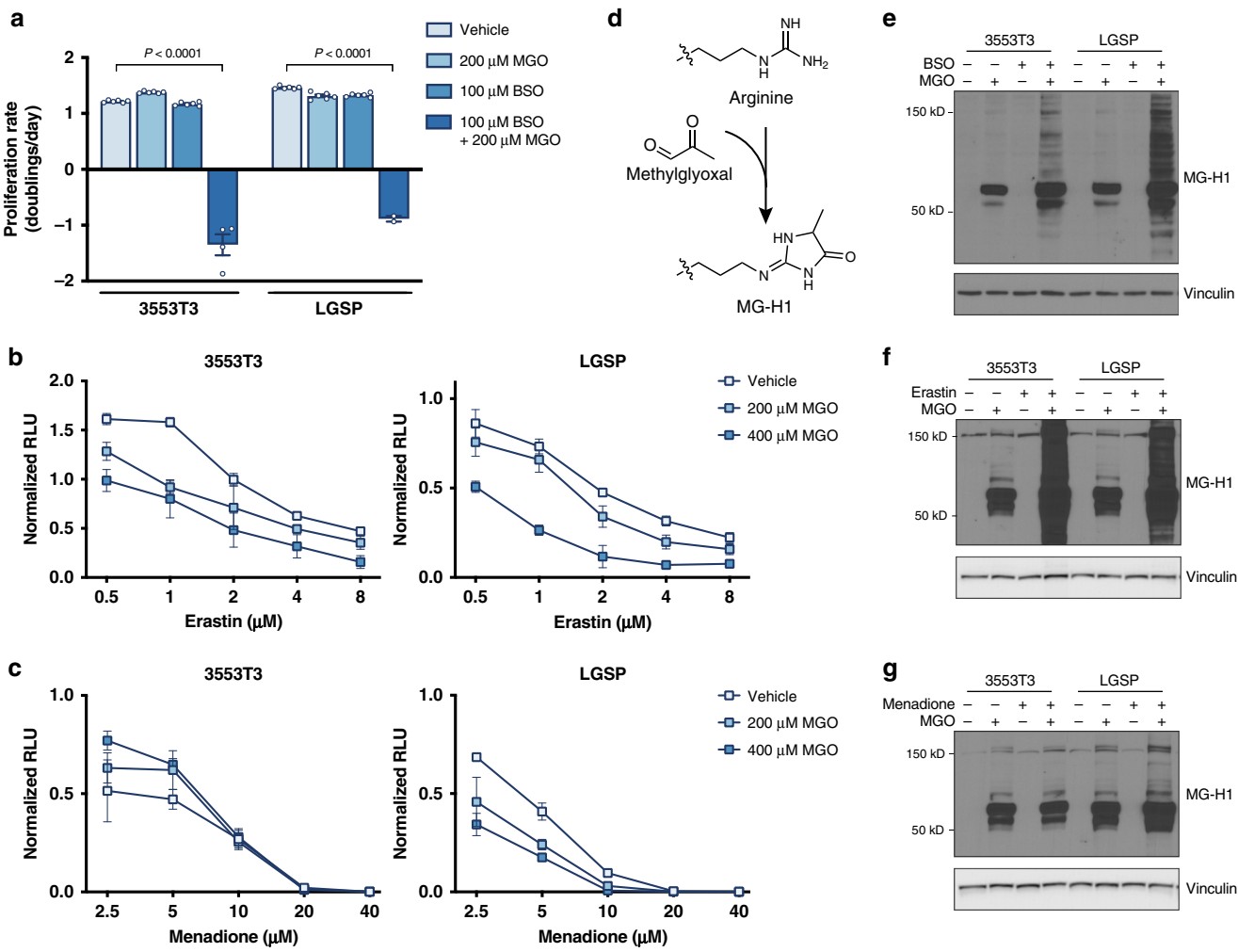

**Fig. 3 Glutathione depletion sensitizes NSCLC cell lines to methylglyoxal. a** Proliferation rate of 3553T3 and LGSP cells cultured in standard media or in media containing the indicated amount of methylglyoxal (MGO) or buthionine sulfoximine (BSO), an inhibitor of glutathione synthesis. Cells treated with BSO had been pre-treated with BSO for 48 h prior to the initiation of the experiment. The $P$ values were calculated by unpaired, two-tailed $t$-test ($n = 6$). **b, c** Effect of the indicated concentration of erastin, which depletes intracellular glutathione by inhibition of system $x_c^-$ (**b**), and menadione, which induces formation of reactive oxygen species (**c**), on cell viability as assessed using the CellTiter-Glo Luminescence Assay in 3553T3 and LGSP cells cultured in standard media or in media containing the indicated doses of methylglyoxal ($n = 3$). **d** Schematic detailing how methylglyoxal can react non-enzymatically with arginine to form the hydroimidazolone MG-H1. **e** Western blot analysis of lysates from 3553T3 and LGSP cells that had been incubated with 200 μM of methylglyoxal, 100 μM of BSO, or both for 24 h using an antibody raised against the MG-H1 epitope. Cells treated with BSO had been pre-treated with BSO for 48 h prior to the initiation of the experiment. **f, g** Western blot analysis of lysates from 3553T3 and LGSP cells to assess accumulation of proteins containing MG-H1 epitopes following 24 h of culture in standard media or in media containing 200 μM methylglyoxal, 10 or 5 μM erastin (3553T3, LGSP respectively), or both as indicated (**f**), or 200 μM methylglyoxal, 2 or 1 μM menadione (3553T3, LGSP respectively), or both as indicated (**g**). Values in panels **a–c** denote mean ± SD.

methylglyoxal-associated adducts. Interestingly, menadione induced accumulation of proteins modified by exogenous methylglyoxal in LGSP, but not 3553T3 cells (Fig. 3g), matching the sensitivity to exogenous methylglyoxal (Fig. 3c). Together, these data suggest that glutathione is required for efficient methylglyoxal detoxification, and confirms that AGE accumulation is associated with decreased lung cancer proliferation and viability.

Since methylglyoxal detoxification requires GSH, we next assessed whether lung cancer cells could adapt to methylglyoxal exposure and glycated biomolecule elevation by upregulating glutathione synthesis. Although exogenous methylglyoxal addition resulted in accumulation of proteins with MG-H1 adducts, expression levels of the catalytic subunit of glutamate-cysteine ligase (GCLC) involved in GSH synthesis remained unchanged (Supplementary Fig. 3b). This suggests that, at least acutely, lung cancer cells may not have a mechanism for increasing

glutathione production as an adaptation to increased methylglyoxal levels.

**Glyoxalase I is required for methylglyoxal detoxification.** To inhibit methylglyoxal detoxification genetically, we utilized CRISPR/Cas9-based genome editing to delete Glo1 from lung cancer cells (Fig. 4a, Supplementary Fig. 4a). Though exogenous methylglyoxal can induce LGSH accumulation in parental lung cancer cells (Supplementary Fig. 4b), levels of intracellular LGSH were dramatically lower in Glo1-deficient cells (Fig. 4b, Supplementary Fig. 4c). Furthermore, Glo1 depletion was found to greatly increase the sensitivity of these cells to exogenous methylglyoxal (Fig. 4c, Supplementary Fig. 4d). Taken together, these data confirm that Glo1 is required for LGSH formation, and that this enzyme represents a major mechanism for methylglyoxal detoxification in lung cancer cell lines.

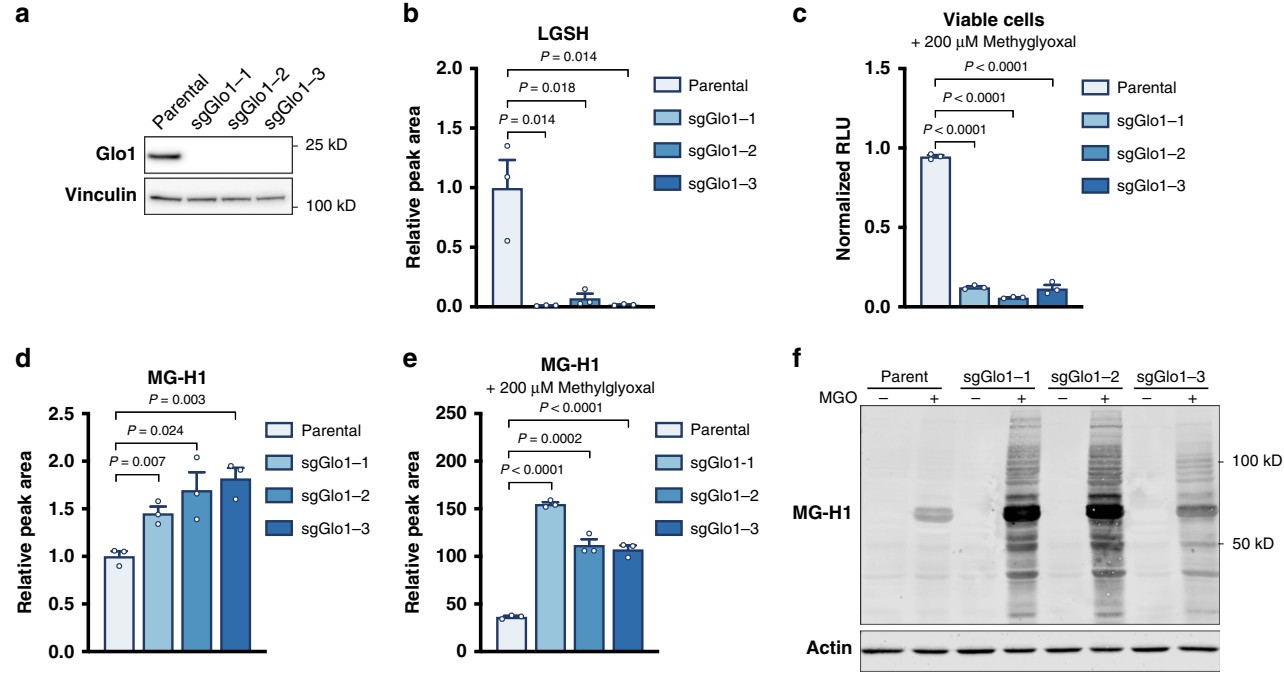

**Fig. 4 Glyoxalase I is required for methylglyoxal detoxification. a** Western blot analysis of glyoxalase I (Glo1) expression in parental 3553T3 lung cancer cells and in three independent clones where the Glo1 gene was disrupted using CRISPR/Cas9 (sgGlo1). **b** Relative levels of lactoylglutathione (LGSH) in 3553T3 parental NSCLC cells or the Glo1-deleted clones described in (**a**) as detected using LCMS. Values shown are the peak area of LGSH normalized to the peak area of GSH ($n = 3$). **c** Relative number of viable 3553T3 parental cells and Glo1-deleted clones as determined by CellTiter-Glo Luminescence Assay following a 48-h incubation with 200 μM methylglyoxal ($n = 3$). **d** LCMS quantification of relative free MG-H1 abundance in 3553T3 parental cells and Glo1-deleted clones. Values are peak area of MG-H1 normalized to peak area of arginine ($n = 3$). **e** LCMS quantification of free MG-H1 abundance in extracts from 3553T3 parental cells and Glo1-deleted clones that were incubation with 200 μM methylglyoxal for 16 h. Values are normalized to MG-H1 peak area in parental lung cancer cell lines incubated without methylglyoxal treatment shown in panel (**d**), which were analyzed in the same LCMS experiment ($n = 3$). **f** Western blot analysis using an antibody raised against the MG-H1 epitope of lysates from 3553T3 parental cells and Glo1-deleted clones that had been treated without or with 200 μM of methylglyoxal for 12 h. Values in **b**–**e** denote mean ± SEM and the $P$ values were calculated by unpaired, two-tailed $t$-test.

We next measured levels of methylglyoxal glycation adducts in cells with and without Glo1 expression. Concentrations of free MG-H1 as measured by LCMS were basally increased in Glo1-deleted cells (Fig. 4d, Supplementary Fig. 4e), and levels of this adduct were further increased when Glo1-deleted cells were incubated with exogenous methylglyoxal (Fig. 4e, Supplementary Fig. 4f, g). Furthermore, increased amounts of proteins containing MG-H1 adducts were also detected by western blot in this context (Fig. 4f, Supplementary Fig. 4h). Together, these data confirm that inhibition of the glyoxalase system in lung cancer cells can promote accumulation of methylglyoxal-derived adducts, and that Glo1 loss can impair cell viability if methylglyoxal is present at high levels.

We next assessed whether NSCLC cells without Glo1 expression were more sensitive to interventions that deplete glutathione than parental cell lines. Lung cancer cells with Glo1 deletion were not found to be differentially sensitive to erastin (Supplementary Fig. 5a) or to menadione (Supplementary Fig. 5b) relative to parental cell lines, suggesting that Glo1 is required to link methylglyoxal detoxification to glutathione availability.

**Glyoxalase I supports NSCLC proliferation under hypoxic conditions**. A major difference between cancer cells in culture and in tumors is that cells in vivo are exposed to reduced levels of oxygen. Hypoxic conditions are known to elevate the rate of glycolysis, and thus methylglyoxal production is expected to be elevated when oxygen is low. Indeed, we found that when cultured in 0.5% oxygen, lung cancer cell lines accumulate higher levels of proteins containing MG-H1 adducts, especially in NSCLC lines that do not express Glo1 (Fig. 5a). Furthermore, we found that hypoxia reduced proliferation of cells that lacked Glo1 (Fig. 5b), suggesting that hypoxic conditions promote accumulation of AGE and reduce cellular fitness.

**Glyoxalase I is required for NSCLC tumor growth**. Since lung cancer cells cultured in hypoxia were more sensitive to methylglyoxal accumulation, we hypothesized that lung tumors might be more sensitive to methylglyoxal under basal conditions. Testing this possibility is important, since the levels of exogenous methylglyoxal needed to cause toxicity in culture are greater than those estimated to exist in plasma in vivo[43]. To determine whether Glo1 expression contributes to lung tumor growth, we implanted parental lung cancer cells and Glo1-deleted cells subcutaneously as mouse allografts. The ability of Glo1-deleted cells to proliferate as allografts was impaired compared to parental NSCLC cancer lines (Fig. 5c). Tumors from Glo1-deleted cells also had increased MG-H1 protein adducts, suggesting that accumulation of methylglyoxal-derived adducts might contribute to reduced tumor growth (Fig. 5d). We repeated the tumor allograft experiment and found that Glo1-deficient tumors exhibit evidence of increased MG-H1 adducts on proteins even at early time points (Supplementary Fig. 6). Taken together, our data argues that Glo1 promotes methylglyoxal detoxification and NSCLC tumor growth.

**Discussion**

Changes in cancer tissue metabolite levels can have important implications for cell physiology. Untargeted metabolomics analysis

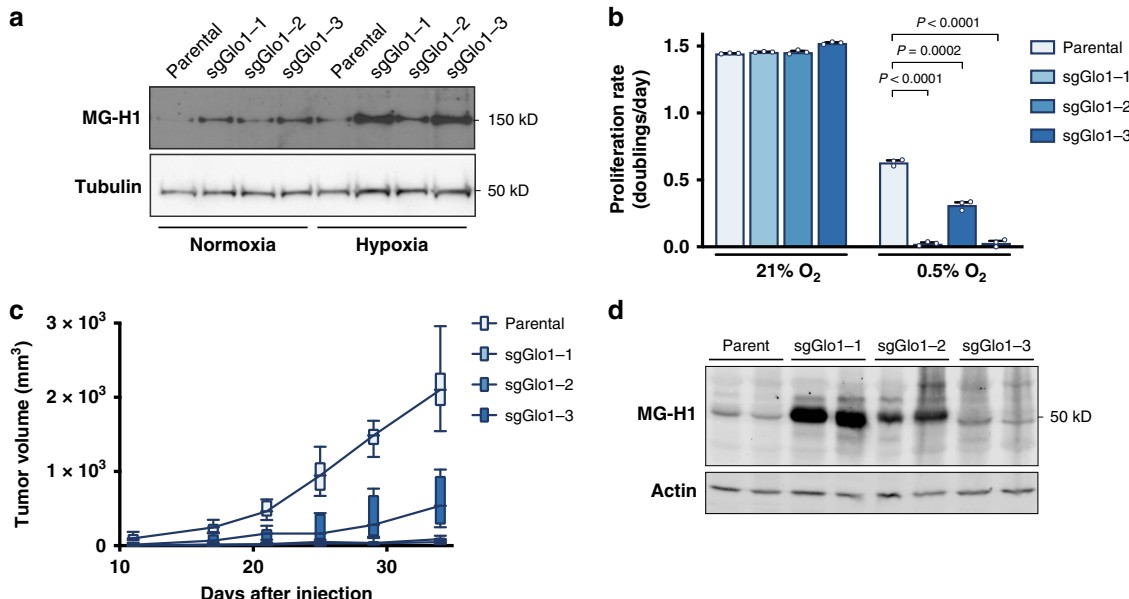

**Fig. 5 Glyoxalase I supports NSCLC proliferation under hypoxic conditions and in vivo. a** Western blot analysis of proteins containing the MG-H1 epitope were performed on lysates collected from 3553T3 parental cells and Glo1-deleted clones that had been cultured under normoxic and hypoxic (0.5% $O_2$) conditions for 72 h. **b** Proliferation rate of 3553T3 parental cells and Glo1-deleted clones when cultured under normoxic and hypoxic (0.5% $O_2$) conditions. Values indicate mean ± SEM, and $P$ values were calculated by unpaired, two-tailed Student's $t$-test ($n = 3$). **c** Tumor growth over time of allografts generated from 3553T3 parental cells and Glo1-deleted clones. The tumor volume for each genotype is depicted as a box and whisker plot for each time point. The difference in tumor growth between parental cell lines and Glo1-deleted clones was significant for every comparison ($P < 0.0001$ by two-way ANOVA; $n = 6$). **d** Western blot analysis using an antibody raised against the MG-H1 epitope of lysates from tumor allografts harvested at the end point of the experiment shown in **d**.

of lung tumors demonstrated that glutathione, which has been shown to be elevated in tumors[19,44,45] and required for malignant transformation[18], accumulates in NSCLC lesions relative to normal lung tissue in mice. Interestingly, levels of the oxidized form of glutathione were not elevated to the same extent in the tumors as reduced glutathione, indicating that there could be an increase in GSH relative to GSSG in NSCLC. This finding is consistent with emerging data that suggests that some tumors are in a more reduced state than normal tissues[46,47], and that this reduced state can be limiting for tumor growth[48].

NSCLC tumors accumulate GSH-electrophile conjugates, including those involving the reactive electrophile methylglyoxal. Glutathione is an obligatory substrate of Glo1, and thus is required for methylglyoxal detoxification. Indeed, we show that agents that deplete intracellular glutathione increase sensitivity to exogenous methylglyoxal and promote accumulation of methylglyoxal glycation adducts. Though high glutathione levels promote methylglyoxal detoxification, it is not expected that increased methylglyoxal production would net consume GSH to a substantial degree. First, glutathione is present at millimolar concentrations in cells, whereas methylglyoxal levels are in the low micromolar range[8]. Second, the enzyme Glo2 can catalyze the conversion of LGSH to D-lactate and regenerate reduced glutathione such that methylglyoxal detoxification does not net consume reduced glutathione. Reduced Glo2 activity could contribute to higher LGSH levels in cells; however, we show experimentally that methylglyoxal is formed as a byproduct of glucose metabolism in NSCLC. High-glucose conditions have been shown to induce accumulation of methylglyoxal adducts for cells in culture[49,50], and LGSH accumulation in NSCLC correlates with increased glucose uptake. This, along with the high abundance of reduced glutathione, is consistent with glutathione playing a role as a scavenger for reactive metabolites, such as methylglyoxal, in some cancers, and exploiting this consequence of increased glucose metabolism could be a strategy to target FDG-PET avid cancers.

The reactive nature of methylglyoxal makes direct measurement of this metabolite difficult, and we relied on LGSH accumulation as a proxy for methylglyoxal production when glyoxalase I is expressed. We find that LGSH accumulates when methylglyoxal is added to lung cancer cells, that LGSH levels are decreased when glycolysis is reduced, and that LGSH levels correlate with [18]F-FDG uptake in human lung tumors. The finding that MG-H1 adducts also track with LGSH levels supports the hypothesis that methylglyoxal production is elevated in the lung cancer cells and tissues examined in the study. However, changes in LGSH metabolism by Glo2 can also affect levels, and thus direct measurement of methylglyoxal could be helpful to further assess the production rate of this reactive metabolite.

Glo1 represents an important mechanism for methylglyoxal detoxification, and has been shown to be elevated in a diverse set of cancer contexts[51,52]. Glo1 ablation suppresses NSCLC tumor growth in vivo, which is in agreement with published studies performed for colon cancer[53], hepatocellular carcinoma[54], and leukemia[55]. We did not observe increased accumulation of AGEs progressively over time in NSCLC allografts that did not express Glo1, which may reflect that these tumors are no longer proliferating or that proteins with methylglyoxal adducts might be degraded. However, we found that tumors derived from parental NSCLC cell lines exhibit higher levels of Glo1 expression at later time points, and increased Glo1 expression was found to correspond to lower levels of methylglyoxal adducts. This could suggest that Glo1 is upregulated during tumor progression or that there is positive selection for increased capacity for methylglyoxal detoxification in tumors.

Taken more broadly, the findings presented in this study suggest that production of other reactive byproducts of cancer cell metabolism, including reactive oxygen species (ROS), formaldehyde, fumarate, and byproducts of lipid peroxidation could represent a targetable vulnerability of tumor cells[11]. That is,

targeting pathways that detoxify reactive metabolites may be an approach for treating tumors by potentiating, rather than repressing, a cancer metabolism phenotype. Though therapies that induce excessive accumulation of reactive metabolites could impair tumor progression, a potential pitfall of this approach is that accumulation of reactive metabolites at low levels can also damage normal cells and may contribute to malignancy. ROS has been shown to activate oncogenic signaling pathways and contribute to genomic instability[56–59]. Endogenously produced formaldehyde can promote systemic DNA damage and malignant transformation[60]. Low doses of methylglyoxal can enhance tumorigenesis even when higher concentrations inhibit tumor progression in the same tissue[61], and reduced expression of Glo1 has been reported to promote breast cancer growth and invasion[62,63]. Understanding the mechanisms by which reactive metabolites promote malignancy or induce toxicity will provide insight into the cases in which elevated production of reactive metabolites represents a strategy to treat cancer.

## Methods

**Mouse models.** For experiments involving the Kras(LA2) mouse lung cancer model[15], 4- to 6-month-old male mice of mixed 129/Sv and C57Bl6 genetic background were used. For experiments involving the Kras$^{LSL-G12D}$;Trp53$^{flox/flox}$ (KP) mouse lung cancer model, 2- to 6-month-old male mice of mixed 129/Sv and C57Bl6 genetic background were used. Lesions in the KP model were initiated by intratracheal delivery of $2 \times 10^7$ PFU of Cre-recombinase (Gene Transfer Vector Core, University of Iowa) adenovirus. For allograft experiments, $1 \times 10^5$ cancer cells derived from the KP mouse model were suspended in 100 μL PBS and injected into the flanks of 4- to 6-week-old male nu/nu Nude mice (Charles River Laboratories, 088). Tumor volume was measured by caliper in two dimensions, and volumes were calculated using the equation $V = (\pi/6)LW^2$, where $L$ is the longer of the two measured dimensions and $W$ is the shorter of the two measured dimensions. Mice were killed after the volume of the first tumor was measured to be >1 cm$^3$ or if recommended by the veterinary staff. All animal work was approved and was compliant with ethical regulations provided by the Massachusetts Institute of Technology Committee on Animal Care.

**Tracing $^{13}$C-Glucose fate in mouse tissues.** Glucose fate was assessed in mouse tissues of conscious mice bearing NSCLC tumors[13]. Catheters were surgically implanted into the jugular veins of tumor bearing animals three days prior to the infusion experiment. The day of the experiment, animals were fasted for six hours prior to [U-$^{13}$C$_6$]glucose infusion for six hours at a rate of 30 mg/kg/min into conscious, free-moving animals. Animals were terminally anesthetized with 120 mg/kg sodium pentobarbital and tumors rapidly collected and frozen utilizing a BioSqueezer (Bio Spec Products Inc.) that had been pre-cooled in liquid nitrogen.

**Metabolite analysis of material from human subjects.** NSCLC patients were enrolled in a protocol approved by the Institutional Review Board at UT-Southwestern Medical Center after obtaining informed consent[64]. Patients were considered eligible for the study if they had solitary pulmonary masses measuring 1 cm or more in diameter. Standard surgical procedures were followed, and the majority were robotic lobectomies. Based on pre-operative imaging and gross inspection at resection, viable fragments of tumor and lung were sampled. Tissue fragments were washed in ice-cold saline and immediately frozen in liquid nitrogen.

Data were acquired with a QExactive HF-X hybrid quadrupole-orbitrap mass spectrometer coupled to a Thermo Scientific Vanquish Flex Ultra-High Performance Liquid Chromatography system (Bremen, Germany). Separation of metabolites was achieved using a Millipore-Sigma SeQuant ZIC-pHILIC column 2.1 × 150 mm (St. Louis, MO) with a binary solvent system. Solvent A consisted of 10 mM ammonium acetate in water which was brought to pH 9.8 with ammonium hydroxide; solvent B was acetonitrile. The gradient began with an initial composition of 90% B which was linearly decreased to 30% in 15 min. Solvent B was held at 30% until 18 min and then brought back to 90% from 18 min to 19 min. The column was re-equilibrated until 27 min. Column temperature was maintained at 25 °C and the flow rate remained constant at 0.25 mL/minute.

HRMS data were acquired with two different methods. Spectra from individual samples were collected with a high-resolution full scan method alternating between both positive and negative polarities. The resolving power of each polarity was set to 60,000 FWHM with a mass range of 80–1200 Daltons; the AGC target was set to $1 \times 10^6$ with a maximum inject time of 100 ms. For the relative quantitation of metabolites, chromatographic peaks from these scans were identified and integrated with a 5 ppm mass tolerance. High-confidence identification of metabolites was achieved with a data-dependent high-resolution tandem mass spectrometry analysis of a pooled sample made from an equal mixture of all

individual samples. Pooled samples were run alongside individual samples to insure chromatographic consistency. For ddHRMS/MS methods, precursor ion scans were collected with a resolving power of 60,000 FWHM with a mass range of 80–1200 Daltons. The AGC target value was set to $1 \times 10^6$ with a maximum injection time of 100 ms. Product ion spectra were collected at a resolving power of 15,000 FWHM without a fixed mass range. The AGC target value was set to $2 \times 10^5$ with a maximum injection time of 150 ms. Data-dependent parameters were set to acquire production ion spectra on the top 10 ions with a dynamic exclusion of 30 s and a mass tolerance of 5 ppm. Isotope exclusion was turned on and a stepped normalized collision energy applied for fragmentation (NCE = 30, 50, 70). These settings remained the same in both polarities for data-dependent acquisition.

**Tumor metabolite extraction.** In all, 10–40 mg tissue fragment was homogenized cryogenically (CryoMill, Retsch). Metabolites were extracted using ice-cold HPLC grade methanol, water, and chloroform in a 6:4:3 ratio. Samples were vortexed at 4 °C for 10 min, and then centrifuged at 4 °C degrees for 10 min to separate the aqueous and organic layers. The top aqueous layer was collected and dried under nitrogen gas for subsequent analysis by mass-spectrometry.

**Untargeted metabolomic analysis.** For untargeted metabolomics analysis, data were acquired using an LCMS system comprised of a Nexera X2 U-HPLC (Shimadzu, Marlborough, MA) coupled to a Q-Exactive hybrid quadrupole orbitrap mass spectrometer (Thermo Fisher Scientific; Waltham, MA). Negative ion mode analyses of polar metabolites were achieved using a HILIC method under basic conditions.

Briefly, dried metabolites extracts were resuspended according to initial tissue wet weight with 80% methanol containing [$^{15}$N]inosine, thymine-d4, and glycocholate-d4 as internal standards (Cambridge Isotope Laboratories; Andover, MA). The samples were centrifuged (10 min, $9000 \times g$, 4 °C) and the supernatants were injected directly onto a 150 × 2.0 mm Luna NH2 column (Phenomenex; Torrance, CA). Samples were eluted at a flow rate of 400 μl/min with initial conditions of 10% mobile phase A (20 mM ammonium acetate and 20 mM ammonium hydroxide in water) and 90% mobile phase B (10 mM ammonium hydroxide in 75:25 vol/vol acetonitrile/methanol) followed by a 10 min linear gradient to 100% mobile phase A. MS full scan data were acquired over m/z 70–750. The ionization source voltage is −3.0 kV and the source temperature is 325 °C. Spectra were processed utilizing Progenesis QI (Nonlinear Dynamics) to identify metabolites that differentially accumulate in lung and tumors, and peaks with fewer than an average of $5 \times 10^4$ ion counts per group were excluded from the analysis. To identify GSH-electrophile conjugates that were elevated in NSCLC tumors relative to normal lung, the human metabolome database[65] was used to identify peaks in the untargeted metabolomics data set with masses corresponding to annotated conjugates.

**Targeted mass spectrometry to measure metabolites.** For metabolite measurement of cells in culture, $1 \times 10^6$ cells were seeded in 10 cm tissue culture dishes. After 24 h, cells were washed three times with PBS and media was changed to 5 mL of the appropriate treatment condition. For isotope labeling experiments, the cells were incubated in and changed to media containing 25 mM [U-$^{13}$C$_6$]glucose or 400 μM [U-$^{13}$C$_2$]glycine. After 12 h, cells were washed rapidly with ice-cold blood bank saline and lysed on the dish with 1 mL of ice-cold 80% HPLC grade methanol in HPLC grade water. Next, samples were vortexed at 4 °C for 10 min, and then centrifuged at 4 °C degrees for 10 min to precipitate the protein. The protein of each sample was quantified using BCA Protein Assay (Pierce 23225). The supernatant was collected, dried under nitrogen gas, and resuspended in 100 μL of 80% HPLC grade methanol in HPLC grade water for subsequent analysis by mass-spectrometry.

Metabolite profiling was conducted on a QExactive bench top orbitrap mass spectrometer equipped with an Ion Max source and a HESI II probe, which was coupled to a Dionex UltiMate 3000 HPLC system (Thermo Fisher Scientific, San Jose, CA). External mass calibration was performed using the standard calibration mixture every 7 days. For each sample 4 μL were injected onto a SeQuant® ZIC®-pHILIC 150 × 2.1 mm analytical column equipped with a 2.1 × 20 mm guard column (both 5 mm particle size; EMD Millipore). Buffer A was 20 mM ammonium carbonate, 0.1% ammonium hydroxide; Buffer B was acetonitrile. The column oven and autosampler tray were held at 25 °C and 4 °C, respectively. The chromatographic gradient was run at a flow rate of 0.150 mL/min as follows: 0–20 min: linear gradient from 80–20% B; 20–20.5 min: linear gradient form 20–80% B; 20.5–28 min: hold at 80% B. The mass spectrometer was operated in full-scan, polarity-switching mode, with the spray voltage set to 3.0 kV, the heated capillary held at 275 °C, and the HESI probe held at 350 °C. The sheath gas flow was set to 40 units, the auxiliary gas flow was set to 15 units, and the sweep gas flow was set to 1 unit. MS data acquisition was performed in a range of m/z = 70–1000, with the resolution set at 70,000, the AGC target at $1 \times 10^6$, and the maximum injection time (Max IT) at 20 ms. For detection of metabolites associated with methylglyoxal, targeted selected ion monitoring (tSIM) scans in positive mode were included. The isolation window was set at 1.0 m/z and tSIM scans were centered at m/z = 229.1295 (Positive Mode) for hydroimidazolone (MG-H1) and m/z = 380.1122 (Positive Mode) and 378.0966 (Negative Mode) for lactoylglutathione

(LGSH). For all tSIM scans, the resolution was set at 70,000, the AGC target was $1 \times 10^5$, and the max IT was 250 msec. MS/MS spectra were collected for LGSH using parallel reaction monitoring mode (PRM) in positive mode with a precursor ion of 320.1122 m/z, an MS/MS resolution of 17,500 and AGC target of $1 \times 10^5$, a maximum IT of 100 ms, and stepped-wise collision energies of 10, 25 and 35.

**Cell culture**. Cell lines were generated from tumors arising from the KP lung cancer model[13]. KP lung lesions were digested in collagenase IV/dispase/trypsin for 30 min at 37 °C. All cell lines were cultured in DMEM (Corning 10-013-CV) supplemented with 10% fetal bovine serum. For all experiments, media was changed to DMEM without pyruvate (Corning 10-017-CV) supplemented with 10% dialyzed fetal bovine serum and the treatment condition as indicated.

To assess cell proliferation under hypoxic conditions, $2 \times 10^4$ cells were seeded and permitted to adhere overnight, then placed into a hypoxia chamber (Baker, InvivO$_2$) that had been previously equilibrated to 0.5% oxygen. After 72 h, cell counts were determined as described below. To assess levels of MG-H1 under hypoxic conditions, cells were harvested for western blot after 72 h.

**Cell proliferation**. For proliferation experiments, $2 \times 10^4$ cells were seeded replicate 6-well dishes. To deplete intracellular glutathione prior to the initiation of the experiment, half the wells were seeded in media containing 200 µM buthionine sulfoximine. After 24 h, one dish was counted to determine the starting cell number prior to treatment. All remaining dishes were washed three times in phosphate buffered saline (PBS), and 4 mL of media containing the indicated treatment was added to cells. Four days following the treatment, cell counts were determined. Cell counts were determined either by using a Cellometer Auto T4 Plus Cell Counter (Nexcelom Bioscience) or quantified with the sulforhodamine B (SRB) assay.

**Sulforhodamine B assay**. Cells were fixed by adding trichloroacetic acid (TCA) then incubated at 4 °C for at least one hour. Fixed cells were washed with deionized water and then stained with 0.057% sulforhodamine B in 1% acetic acid for 30 min. Following three washes with 1% acetic acid, plates were air-dried at room temperature. In all, 1 mL of 10 mM Tris [pH 10.5] was added to each well to solubilize SRB dye, and the OD at 510 nm was measured in a microplate reader (Tecan Infinite M200Pro).

**Cell titer Glo experiments**. To measure cell growth, $2.5 \times 10^3$ cells were seeded in 96-well dishes. After overnight attachment, cells were washed with PBS then treated with media containing erastin or menadione. Relative cell numbers were determined following 48 h treatment using CellTiter-Glo® Luminescent Cell Viability Assay (Promega G7572) according to manufacturer's recommendations.

**CRISPR/Cas9-mediated disruption of *Glo1***. E-CRISP (e-crisp.org) was used to design sgRNA targeting *Glo1* (F 5′-CACCGCTGTCACCCACCTTGGTGCTGTT T-3′; R 5′-AAACAGCACCAAGGTGGGTGACAGCGGTG-3′). The sgRNA was cloned into lentiCRISPR v2 (Addgene, 52961). Isogenic clones were selected for comparison of CRISPR cell lines.

**Calculation of methylglyoxal isotopomer distribution**. The isotopomer distribution of methylglyoxal was calculated from the isotopomer distribution of lactoylglutathione and reduced glutathione as the solution to the linear system $Ax = b$ with:

$$\text{GSH } A = \begin{bmatrix} m+0 & 0 & 0 & 0 \\ m+1 & m+0 & 0 & 0 \\ m+2 & m+1 & m+0 & 0 \\ m+3 & m+2 & m+1 & m+0 \\ \vdots & \vdots & \vdots & \vdots \\ m+10 & m+9 & m+8 & m+7 \\ 0 & m+10 & m+9 & m+8 \\ 0 & 0 & m+10 & m+9 \\ 0 & 0 & 0 & m+10 \end{bmatrix}$$

$$\text{LGSH } b = \begin{bmatrix} m+0 \\ m+1 \\ \vdots \\ m+13 \end{bmatrix}$$

$$\text{Methylglyoxal } x = \begin{bmatrix} m+0 \\ m+1 \\ m+2 \\ m+3 \end{bmatrix}$$

**FDG-PET scans**. Imaging of tumors using combined positron emission tomography (PET) and x-ray computerized tomography (CT) was used to image the distribution of the radiolabeled tracer 2-deoxy-2-[$^{18}$F]fluoro-D-glucose (FDG) in NSCLC patients[64]. PET-CT exams were performed using a Siemens Biograph 64 (2007) PET/CT Scanner (Siemens Healthcare). Patients were injected with 5.7 Mbq/kg FDG intravenously, with a minimum dose of 370 Mbq and maximum dose of 740 MBq. Images were acquired ~1 h following FDG dosing. The parameters for the non-contrast CT images were 120 kVp, fixed 200 mAS, 3 mm slice thickness in the head and neck and 5 mm slice thickness of the body, pitch 0.8 s and rotation time 0.5 s. These images were concurrently acquired for attenuation correction and anatomic correlation with PET images. The data was analyzed using a Syngo.Via station using an automated gradient-based segmentation method, and standardized uptake values (SUV) were calculated with the following formula: SUV = (maximum tissue activity of FDG)/(injected dose of FDG × patient body weight).

**Western blot analysis**. Cells were washed with cold PBS and lysed with cold RIPA buffer containing cOmplete Mini protease inhibitor (Roche 11836170001). Protein concentration was quantified by BCA Protein Assay (Pierce 23225) with BSA as a standard. Lysates were resolved by SDS-PAGE using standard techniques. Proteins were transferred onto nitrocellulose membranes using the iBlot2 Dry Blotting System (Thermo Fisher, IB21001, IB23001). For cells cultured in hypoxia, cells were washed with PBS and lysed in urea lysis buffer containing protease and phosphatase inhibitors (6.5 M urea, 50 mM Tris [pH 7.5], 1 mM EGTA, 2 mM DTT, 1 mM PMSF), and protein concentration was quantified by Pierce 600 nm Protein Assay (Pierce 22660). Proteins were detected with the following primary antibodies: anti-Methylglyoxal (1:1000; Cell Biolabs, STA-011), anti-Glyoxalase I (1:1000; Novus Biologicals, 19015), anti-γ-glutamylcysteiene glutamylcysteine synthetase heavy subunit (1:200, Santa Cruz Biotechnology sc-22755), anti-Vinculin (1:10,000; Sigma, V9131), and anti-α-Tubulin (1:1000; Abcam, ab4074), anti-β-Actin (1:1000; Cell Signaling Technology 8457). The secondary antibodies used were IR680LT dye conjugated anti-rabbit IgG (1:10,000; Licor Biosciences 925-68021), IRDye 800CW conjugated anti-mouse IgG (1:10,000; Licor Biosciences 925–32210), HRP-conjugated anti-rabbit IgG (Millipore 12–348), and HRP-conjugated anti-mouse IgG (Millipore 12–349). Unprocessed western blot images are included in the Supplementary Figs. 7 and 8.

**Statistical analysis**. Data are presented as mean ± SEM and sample size (n) indicates replicates from a single representative experiment, unless otherwise indicated in the figure legends.

## Data availability

All data generated to support the findings of this study are available from the corresponding author upon reasonable request.

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

## Acknowledgements

We thank all members of the Vander Heiden lab, Isaac Harris, and Robert J. Downey for thoughtful discussion and advice. We also thank Genya Frenkel and Kendall Condon for assistance with the hypoxia experiment and Lukas Murmann for help with linear algebra. This work was supported by the Ludwig Center for Molecular Oncology Fund (A.L.), NSF GRFP DGE-1122374 (A.L.; K.L.A.; S.M.D.) and T32GM007287 (A.L.; K.L.A.; S.M.D.; A.M.H). E.F. was supported by United States Department of Defense Peer-Reviewed Medical Research Program grant W81XWH-15-1. R.J.D. acknowledges support from the Howard Hughes Medical Institute, a Cancer Prevention and Research Institute of Texas RP160089, and the National Cancer Institute (R35CA220449). M.G.V.H. acknowledges support from a Faculty Scholar grant from the Howard Hughes Medical Institute, SU2C, the Lustgarten Foundation, the MIT Center for Precision Cancer Medicine, the Ludwig Center at MIT, and the NIH (R01CA201276, R01CA168653, and P30CA14051).

## Author contributions

A.L. and M.G.V.H. designed the study and A.L. and K.L.A. performed most experiments. S.M.D. and C.B.C performed the untargeted LCMS metabolomics and C.A.L. and S.C. analyzed the untargeted LCMS metabolomics. C.A.L., S.C., and E.F. performed the targeted LCMS experiments. A.L. and S.M.D. performed the mouse $^{13}$C glucose infusions. A.L. and A.M.H. performed the linear algebra calculation. B.F., L.G.Z., T.P.M., and R.J.D. measured lactoylglutathione levels in human tumors. A.L. and M.G.V.H. wrote the manuscript with input from all authors.

## Competing interests

M.G.V.H. and R.J.D. disclose that they are scientific advisory members for Agios Pharmaceuticals and M.G.V.H is a scientific advisory board member for Aeglea Biotherapeutics, Auron Therapeutics, and iTeos Therapeutics. Since the completion of this work A.L. has started a position at a Flagship Pioneering biotechnology start-up company. All other authors declare no competing interests.
