## [Peer Review File · Nature Communications]

REVIEWERS' COMMENTS:

Reviewer #4 (Remarks to the Author):

I am of the opinion that the authors have appropriately responded to the previous reviews and see no reason why this paper should not be published in Nat Comms. My only reservation is the removal of the data of the measurements of methylglyoxal. I understand this was in response to reviewers' comments, but if the measurements were obtained using reliable and accepted methods, which it seems they were, and they are robust and reproducible, then it seems to me it would be of value to the field to include these data. After all this is what the MS is largely about. Up the authors and editor, but I wouldn't remove data to placate reviewers. Unless of course the authors have any reservations about that data set or the methods used to derive them

REVIEWERS' COMMENTS: Reviewer #4 (Remarks to the Author): I am of the opinion that the authors have appropriately responded to the previous reviews and see no reason why this paper should not be published in Nat Comms. My only reservation is the removal of the data of the measurements of methylglyoxal. I understand this was in response to reviewers' comments, but if the measurements were obtained using reliable and accepted methods, which it seems they were, and they are robust and reproducible, then it seems to me it would be of value to the field to include these data. After all this is what the MS is largely about. Up the authors and editor, but I wouldn't remove data to placate reviewers. Unless of course the authors have any reservations about that data set or the methods used to derive them

We thank this Reviewer for their kind words and for their time reviewing the manuscript. We chose to remove the original methylglyoxal data because efforts to validate these were inconsistent from a technical perspective, and the availability of human patient samples precluded efforts to optimize detection methods in a timely fashion. Because we cannot be fully confident in the existing measurements we prefer not to include that methylglyoxal data in this publication. Work is ongoing on a separate derivatization method to reliably measure methylglyoxal, and we hope to use this in future studies to reliably measure methylglyoxal in cells and tissues.